# Effects of Different Bedding Materials on Production Performance, Lying Behavior and Welfare of Dairy Buffaloes

**DOI:** 10.3390/ani13050842

**Published:** 2023-02-25

**Authors:** Kaifeng Niu, Zhigao An, Zhiqiu Yao, Chao Chen, Liguo Yang, Jiajun Xiong

**Affiliations:** 1Key Laboratory of Animal Genetics, Breeding and Reproduction, Ministry of Education, College of Animal Science and Technology, Huazhong Agricultural University, Wuhan 430070, China; 2International Joint Research Centre for Animal Genetics, Breeding and Reproduction (IJRCAGBR), Huazhong Agricultural University, Wuhan 430070, China; 3Hubei Province’s Engineering Research Center in Buffalo Breeding and Products, Wuhan 430070, China

**Keywords:** bedding material, buffalo, behaviororistics, milk yeild, animal welfare

## Abstract

**Simple Summary:**

Different bedding materials have important effects on the growth, performance, lying behavior and animal welfare of buffaloes. In this study, we investigated the effects of two bedding materials on lying behavior, production performance and animal welfare of dairy buffaloes. Our results indicated that FMB (fermented manure bedding) promoted the performance and welfare of buffaloes. The results showed that the application of FMB improved the lying behavior of buffaloes, the average daily lying time (ADLT) of buffaloes in FMB increased by 58 min compared to those in CB and the average daily standing time (ADST) decreased by 30 min. The results indicated that the average daily milk yield of buffaloes in FMB increased by 5.78% compared to buffaloes in CB. The results showed that the application of FMB improved the welfare of buffaloes. In summary, the application of FMB has significantly modified the lying behavior, production performance and welfare of buffaloes and significantly reduced the cost of bedding material.

**Abstract:**

Different bedding materials have important effects on the behavioristics, production performance and welfare of buffalo. This study aimed to compare the effects of two bedding materials on lying behavior, production performance and animal welfare of dairy buffaloes. More than 40 multiparous lactating buffaloes were randomly divided into two groups, which were raised on fermented manure bedding (FMB) and chaff bedding (CB). The results showed that the application of FMB improved the lying behavior of buffaloes, the average daily lying time (ADLT) of buffaloes in FMB increased by 58 min compared to those in CB, with a significant difference (*p* < 0.05); the average daily standing time (ADST) decreased by 30 min, with a significant difference (*p* < 0.05); and the buffalo comfort index (BCI) increased, but the difference was not significant (*p* > 0.05). The average daily milk yield of buffaloes in FMB increased by 5.78% compared to buffaloes in CB. The application of FMB improved the hygiene of buffaloes. The locomotion score and hock lesion score were not significantly different between the two groups and all buffaloes did not show moderate and severe lameness. The price of FMB was calculated to be 46% of CB, which greatly reduced the cost of bedding material. In summary, FMB has significantly improved the lying behavior, production performance and welfare of buffaloes and significantly reduce the cost of bedding material.

## 1. Introduction

The most used bedding materials in current barns are wood chips and sand [1,2]. Other materials, including straw and peanut shells, are also often used [3,4]. A study noted that the increasing demand for bedding materials has led to the increasing price of common bedding materials [5]. Therefore, recycled manure bedding materials are increasingly preferred by farmers [6]. The fermented manure bedding can be produced by ectopic fermentation with buffalo manure as raw material [7]. At present, there are three main approaches to the cattle manure as bedding material: the solid–liquid separation direct utilization model [8], anaerobic fermentation bedding production model [9] and aerobic fermentation bedding production model [10].

Research shows that bedding plays a key role in improving bovine welfare and lying behavior [11,12]. Beef kept on rubber mats have longer lying time, significantly better body hygiene and less joint damage compared to concrete floors [13]. A study compared the effects of three bedding materials, concrete, soft rubber and sand, on the lying behavior of cows. The results showed that cows spent the longest time lying on the rubber bedding [14]. Another study compared the effects of peanut shell and rice husk on the lying behavior of dairy cows, and the results showed that dairy cows had more lying time and lying frequency on rice husk (Li P et al., 2021) [15]. Generally, the milk yield of cows is positively correlated with the lying time, and the longer the lying time, the higher the milk yield [16,17]. Cows spent more time lying on the bed when the bedding materials were dry and soft [18]. A study noted that, after transferring cows housed in concrete bedding to barns with deep fermented manure bedding, milk production per dairy cow increased by an average of 13.3% [19]. A study compared the effects of recycled manure, new sand, recycle sand and foam-core mattress bedding material on the milk yield of dairy cows. The results showed that dairy cows raised on new sand and recycled manure bedding had a higher milk yield [20].

Different bedding materials have an important impact on welfare indices, such as the hoof health and hygiene of cattle [21]. Fernández et al. [22] concluded that compost-bedded packs significantly improved the welfare and comfort of cows. Housing bulls on deep fermented manure bedding significantly improved the hock lesion and injuries and body hygiene compared to the concrete floors [23]. Moreover, deep fermented manure bedding was more effective in protecting cattle from joint injury and less risk of cattle limb and hoof disease [24]. Research showed that the use of deep recycled manure bedding in free-stall barns reduced the prevalence of lameness and joint wear, and it can improve body hygiene in dairy cattle [25].

Therefore, the aim of this study was to compare the effects of chaff bedding (CB) and fermented manure bedding (FMB) on buffalo lying behavior, milk production and animal welfare. By recording and observing dairy buffaloes for up to one year, we collected a series of indicators, such as lying behavior, milk production and animal welfare of buffaloes to evaluate the application prospect of fermented manure bedding.

## 2. Materials and Methods

This study was conducted from December 2020 to December 2021 in Jinniu Animal Husbandry Co LTD, Jingmen city, Hubei province, China (Longitude: 112.07438; latitude: 30.72365). The protocol of this experiment was approved by the Scientific Ethics Committee of Huazhong Agricultural University (HZAUBU-2020-0003) and the animal trial was conducted in accordance with the National Institute of Health Guidelines for the Care and Use of Experiment Animals (Beijing, China).

### 2.1. Manufacture of Fermented Manure Bedding

The process of using buffalo manure as a raw material and harmless ectopic fermentation treatment as bedding material is shown in Figure 1. Firstly, fresh buffalo manure and rice chaff were collected. The moisture content of the chaff was less than 5%, and the moisture content of the mixture was adjusted to approximately 60% by adding chaff into the buffalo manure and mixing it evenly with a forklift. Then, the microbial agents were evenly sprinkled at 125 g/m^3^ and mixed to build a strip chopped heap with a length of 8 m, a width of 6 m and a height of 1.5 m (The microbial agent mainly contained bacillus subtilis DK068, lactic acid bacteria, yeast MX0126, cellulase and lignin enzyme, with viable bacteria count > 1.0 × 10^10^ CFU/g). After that, the temperature of the pile will start to rise. After 7–10 days, the temperature will reach 55 °C, turn the pile with forklift every 4 days thereafter. The high temperature period (>55 °C) can be maintained for about 20 days, and the highest temperature can reach more than 75 °C. The fermentation period is about 40 days. The moisture content of FMB is reduced from 60% to 35–40%. The fermented manure bedding and chaff bedding were laid flat in the stall. FMB was laid in the experimental group and CB in the control group, and the thickness of the bedding was 50 cm.

### 2.2. Animal Management

More than 40 multiparous lactating buffaloes were randomly divided into 2 groups according to age. All the buffaloes in this study were lactating Mediterranean hybrids, and all were in early or early–middle lactation stage. There were 20 lactating buffaloes in FMB group and 21 in CB group from January to June. There were 21 lactating buffaloes in FMB group and 23 in CB group from July to December. The barn is an east–west oriented open barn with a head-to-head double-column interior. The cowshed was a semi-enclosed barn with pens, and the water tank was located outside the barn. The two stalls on the south and north sides of the same barn were selected; each stall was about 35 m long and 10 m wide with an area of about 350 m^2^, and the exercise area of each buffalo was not less than 15 m^2^. The trial period was one year with a full mixed diet twice daily (07:00 and 15:00), ad libitum feeding and watering. Both stalls were supplemented with the appropriate bedding periodically every two months.

### 2.3. Measurement Index

#### 2.3.1. Lying Behavioral Index

Using a combination of cameras and human observation, high-definition cameras were installed at two opposite corners of each stall to continuously observe and count buffaloes throughout the day (0:00–24:00). These HD cameras were connected to a digital video recorder for recording (two cameras were installed at each diagonal of the cattle stall). HD cameras for daytime and nighttime monitoring of buffaloes are shown in Figure 2. Behavioral indicators of buffaloes were observed throughout 2021, recording 12 days per month. The 4th–7th, 14th–17th, and 24th–27th of each month were chosen to record the behavioral indicators of the buffaloes. The main observation and statistical items: lying time, standing time and buffalo comfort index.

Lying: the belly of the body is in contact with the ground, and the body is supported by the ground rather than the hooves and legs.

Standing: the body is supported by at least three legs on the ground.

Buffalo Comfort Index: 1 h after each buffalo is milked, returned and fed, the number of buffalos lying in the stall/total number of buffaloes.

#### 2.3.2. Milk Yield

Buffaloes were milked twice a day, usually once in the morning at 3–4:00 a.m. and once in the afternoon at 15–16:00 p.m. The daily milk yield of each buffalo was collected and recorded.

#### 2.3.3. Animal Welfare Index

During the experiment, the two groups’ body hygiene score, locomotion score and hock lesion score of the buffaloes were scored individually every month by 1 trained observer, 12 times in total.

The buffalo body hygiene score (BHS) was assessed by the amount of dirt on the udder and lower hind legs based on a 5-point scale with 1 = clean and 5 = dirty [26]. Buffalo body hygiene score < 3 is regarded as clean, and ≥3 is regarded as dirty. The average body condition and hygiene score were calculated for each pen for analysis.

Buffaloes were evaluated for lameness using a 5-point locomotion scoring method [27]. Locomotion score (LS) was as follows: 1 = normal locomotion, 2 = imperfect locomotion, 3 = lame, 4 = moderately lame and 5 = severely lame.

The severity of back leg hock lesion (HL) was measured using the 6-point scale scoring system as described previously [28]. Hock lesions were classified as 1 = no lesion, 3 = hair loss (mild lesion) and 6 = swollen hock with hair loss (severe lesion).

### 2.4. Statistical Analysis 

Statistical analysis was performed by SPSS software (SPSS v. 29, SPSS Inc.; Chicago, IL, USA). This study analyzed the production performance, lying behavior indexes and animal welfare indexes of buffaloes and were conducted by two-way ANOVA in SPSS. *p* < 0.05 was used to indicate a significant difference.

## 3. Results

### 3.1. Lying Behavioral Index

The factors affecting the lying time of buffaloes are shown in Table 1. The results showed that different months and bedding materials have an effect on the lying time of buffaloes (*p* < 0.001), and there is an interaction between month and bedding (*p* < 0.01).

The results of the average daily lying time of buffaloes with different bedding are shown in Table 2; the buffalo ADLT was 606 min in the CB group and 664 min in the FMB group, and the FMB group increased the buffaloes’ ADLT by 58 min compared with the CB group. The ADLT of buffaloes raised in the FMB group was higher than that in the CB group in July, but the difference was not statistically significant (*p* > 0.05). The difference was significant (*p* < 0.01) in the rest of the months.

Buffaloes had the least ADLT in August–September, with only 542 min and 550 min in the CB group and 615 min and 616 min in the FMB group, respectively. The reason should be that at this time the ambient temperature is high, and buffalo are most severely affected by heat stress resulting in a decrease in ADLT. The buffalo had the longest ADLT in April–May when the ambient temperature was optimal, with 633 min and 646 min in the CB group and reaching 722 min and 713 min in the FMB group, respectively.

The factors affecting the standing time of buffaloes are shown in Table 3. The results showed that different months and bedding materials have an effect on the lying time of buffaloes (*p* < 0.001), and there is an interaction between month and bedding (*p* < 0.05).

The results of the average daily standing time of buffaloes with different bedding are shown in Table 4. The ADST was 354 min in the CB group and 324 min in the FMB group, and the FMB group decreased by 30 min compared with the CB group. The ADST of buffaloes raised in the FMB group was lower than that in the CB group in February and June, but the difference was not statistically significant (*p* > 0.05). The difference was significant (*p* < 0.05) in the rest of the months. 

The results showed that buffaloes in the CB group had the longest ADST in August–September, reaching 378 min and 371 min, respectively, while buffaloes in the FMB group had the most ADST in July–September, reaching 341, 340 and 342 min, respectively. The results showed that buffaloes had the least ADST in April–May, with only 346 min and 327 min in the CB group and 307 min and 308 min in the FMB group, respectively.

The results of the BCI of buffaloes with different bedding are shown in Table 5. The BCI (a.m.) of buffaloes was 86.56% in the CB group and 89.45% in the FMB group. The BCI (a.m.) in July was 82.53% in the CB group and 86.91% in the FMB group, with significant differences (*p* < 0.05); the differences were not significant (*p* > 0.05) in other months. The BCI (p.m.) of buffaloes was 85.82% in the CB group and 88.33% in the FMB group. The BCI (p.m.) in October was 85.64% in the CB group and 90.40% in the FMB group, with significant differences (*p* < 0.05); the differences were not significant (*p* > 0.05) in other months.

### 3.2. Milk Yield

The results of the average daily milk yield of buffaloes with different bedding materials are shown in Table 6. The buffalo ADMY was 4.39 kg in the FMB group and 4.15 kg in the CB group, and the difference was significantly. The buffalo ADMY increased by 5.78% in the FMB group compared to the CB group. The FMB group was higher than the CB group in June, but the difference was not significant (*p* > 0.05), and the difference was significant (*p* < 0.05) in other months.

The results showed that the ADMY of lactating buffalo was lowest in July–September, which were 3.8 kg, 3.9 kg and 3.75 kg in the CB group, respectively; the FMB group were 4.05 kg, 4.13 kg and 4.05 kg, respectively. It is speculated that the high ambient temperature in this period, which causes the buffalo to be seriously affected by heat stress, and the reduced lying time resulted in lower milk yield.

### 3.3. Animal Welfare Index

The results of the body hygiene score of buffaloes with different bedding are shown in Table 7, and the body hygiene of the buffalo is dirty when the BHS ≥ 3. The BHS of buffaloes in the FMB group was 1.81 and 1.89 in the CB group, with no significant difference (*p* > 0.05). However, the proportion of dirty buffaloes in the CB group was 29.93%, which was significantly higher than 24.80% in the FMB group (*p* < 0.01).

The results of the locomotion score of buffaloes with different bedding are shown in Table 8. The LS of buffaloes in the FMB group was 1.41 and 1.45 in the CB group, with no significant difference (*p* > 0.05). All buffaloes in both groups did not show moderate and severe lameness during the experiment.

The results of the hock lesion score of buffaloes with different bedding are shown in Table 9. During the experiment, the HLS of buffaloes in the FMB group was 0.87 and 0.90 in the CB group, with no significant difference (*p* > 0.05). All buffaloes in both groups did not show moderate and severe damage to the hock joints.

### 3.4. Costing

The cost of the chaff bedding material was calculated to be 52 USD/ton; the cost of fermented manure bedding materials was 6.67 USD/ton. The area of the two stalls is similar: approximately 350 m^2^ and the thickness of bedding is 50 cm. The cost of chaff bedding spread over the whole stall is approximately USD 3633. The cost of the bedding materials of fermented manure bedding spread over the whole stall is approximately USD 1631, which can save USD 2002 on bedding cost.

## 4. Discussion

### 4.1. Lying Behavior

Cattle spend a lot of time lying on bedding every day; therefore, the comfort level of the bedding material in the stall has a direct impact on the quality of their rest. A study showed that the ADLT for cows on straw bedding beds was 749 min, significantly higher than 678 min on sand bedding beds [28]. Angus bulls raised on rubber bedding had significantly more lying time than those on concrete floors [13]. The cows raised on deep sand bedding had significantly more lying time and lying frequency than those on rubber bedding [29]. The cows raised on dry manure bedding had a 2.2 h higher lying time and 2 times higher lying frequency than the cows on rubber bedding [30]. Fernández et al. [22] and Marcondes’ et al. [19] studies showed that compost bedded pack barns have a positive impact on cows’ performance, animal welfare and comfort. The results of this experiment showed that the ADLT of buffaloes raised in the FMB group was 664 min, and was 606 min in the CB group. The ADLT of buffaloes in the FMB group increased by 58 min compared to the CB group. The difference in ADLT was not significant (*p* > 0.05) in July, but the difference was significant (*p* < 0.05) in the rest of the months. The ADST of buffaloes was 324 min in the FMB group and 354 min in the CB group, and the ADST of buffaloes in the FMB group was reduced by 30 min compared with that in the CB group. The difference in ADST in February and June was not significant (*p* > 0.05), but the rest of the months were significantly different (*p* < 0.05).

Studies have shown that ADLT is shortest in cattle in summer, significantly lower than in spring and autumn [31]. The ADLT of cows during the strong heat stress period in June–August was significantly lower than during the mild heat stress period [32]. Smith et al. [33] found that, under heat stress conditions, ADLT and lying frequency were higher in cows with evaporative pads than in cows without evaporative pads. Studies have shown that the more severe the heat shock, the lower the average daily lying time and frequency of cows. Cow core body temperature rises when lying down and decreases when standing [34]. This is consistent with the results of this study, where buffaloes had the least ADLT and the longest ADST in August–September. In the CB group, ADLT was only 542 min and 550 min, and ADST was 378 min and 371 min. In the FMB group, ADLT was 615 min and 616 min, and ADST was 340 min and 342 min. The reason should be that at this time the ambient temperature is high and buffaloes are most severely affected by heat stress, resulting in a decrease in ADLT and an increase in ADST. In April and May, buffaloes had the most ADLT and the least ADST. The CB groups’ ADLT was 633 min and 646 min, and ADST was 346 min and 327 min. In the FMB group, ADLT reached 722 min and 713 min, and ADST was 307 min and 308 min.

The Cow Comfort Index (CCI) was proposed by Nelson in 1996 [35] and is now widely used as a stall comfort assessment. In general, the maximum value of the comfort index for cows occurs in the morning and 1 h after returning from afternoon milking [36]; a CCI > 85% indicates high cow comfort, and this index can also be applied to buffaloes (Buffalo Comfort Index). In this study, both groups of buffalo BCI (a.m.) were greater than 85%, indicating high comfort level of buffalo. However, the BCI (p.m.) of the CB group was less than 85% in July–September, and the BCI (p.m.) of the FMB group was greater than 85%, indicating that fermented manure bedding can improve the comfort of buffalo. In summary, the application of FMB improved the lying behavior of buffaloes compared to CB bedding.

### 4.2. Milk Yield

A study showed that the milk yield of cows is positively correlated with the lying time, and the longer the lying time, the higher the milk yield [16,17]. Dry and soft bedding in the barn provided an excellent and comfortable environment for cattle to rest and grow [6]. Graunke et al. [37] found that growing holstein cows raised on soft rubber mats increased daily weight gain by 9.09% compared to cows raised on concrete floors. The average daily milk yield of cows raised on recycled manure bedding was extremely significantly higher than that on hardened floors [38]. Milk yield per cow in the barn increased by an average of 13.3% after transferring cows housed in concrete floor barns to barns with deep recycled manure bedding [19]. 

The results of this study showed that the average milk yield of buffaloes in the CB group was 4.15 kg/d and the average milk yield of buffaloes in the FMB group was 4.39 kg/d. The average daily milk yield of buffaloes in the FMB group increased 5.78% compared to buffaloes in the CB group. As the ADLT of buffaloes in the FMB group was 60 min higher than that of the CB group, this was the reason that the milk yield of buffaloes in the FMB group was higher than that of the CB group.

### 4.3. Welfare Indexes

Cattle body hygiene is one of the important animal welfare indexes and it is influenced by several factors, the type of bedding being one of the key factors [21]. Previous studies have shown that cows raised on the deep sand bedding had better hygiene than those in the regular straw bedding [28]; both used recycled cow manure as bedding material, and cattle in the deep bedding group had better body hygiene than the shallow bedding group, but the difference was not significant [25]. Guarín et al. [39] compared the effect of four bedding materials (new sand, recycled sand, deep-bedded manure solids and shallow-bedded manure solids over foam core mattresses) on udder cleanliness of cows. The udder cleanliness of cattle raised on deep-bedded manure solids bedding reached 95%. In this study, the body hygiene score of buffaloes in the FMB group was lower than that in the CB group, the difference was not significant (*p* > 0.05). However, the percentage of dirty buffaloes was 29.93% in the CB group and 24.80% in the FMB group, the difference was significant (*p* < 0.01). The reason might be that the bedding in the FMB group was made from buffalo manure and fermented with microbial agents. Fermented manure bedding contains a large number of beneficial microorganisms. After buffalo treading and plowing to mix fresh manure and urine, it facilitates the microorganisms in the bedding to decompose the organic matter in manure and urine and accelerate the evaporation of water in the bedding.

The locomotion score and hock lesion score can identify the early lameness of cattle, determine the hoof health of cattle and make reference for early management adjustment [27]. It has been shown that cows raised on rubber mats have lower levels of joint damage compared to concrete floors [13]. Similarly, less hock injury and joint wear and healthier limbs and hooves have been reported in pastures with deep sand or deep recycled manure bedding [25,40]. Barrientos et al. [41] surveyed 79 farms, and cows with bedding depths of at least 10 cm had healthier hoof limbs. By testing serum biomarkers of joint damage in cows, the degree of lameness and joint damage was more severe in cows raised on concrete floors than in deep sand bedding beds and deep fermented manure bedding [24].

In this study, the bedding thickness was 50 cm for both the FMB and CB groups, and the results showed that the locomotion score of buffaloes was 1.41 in the FMB group and 1.45 in the CB group, with no significant difference between the two groups (*p* > 0.05). Moreover, all buffaloes did not show moderate or severe lameness during the experiment. No moderate and severe joint injuries were observed in all buffaloes. It indicates that the deep bedding played a cushioning and supporting role for the limb and hoof of cattle, which can effectively protect the limb and hoof of cattle.

## 5. Conclusions

In this study, we compared the effects of two bedding materials, FMB and CB, on lying behavior, performance and animal welfare of dairy buffaloes. The results showed that FMB significantly improved the lying behavior of dairy buffaloes compared with CB and significantly increased the milk production of dairy buffaloes. The application of FMB improved the hygiene welfare of dairy buffaloes. The differences in locomotion scores and hock lesion scores between the two groups were not significant, and all dairy buffaloes showed no moderate and severe joint damage. In addition, the price of FMB was calculated to be 4.74 USD/m^2^ and the price of CB was 10.38 USD/m^2^. The price of FMB was 46% of the price of CB, which greatly reduced the cost of bedding material. The results of this study provide a reference for the rational utilization of buffalo manure resources, reducing bedding costs and achieving the coordinated development of cattle breeding and ecological environment.

## Figures and Tables

**Figure 1 animals-13-00842-f001:**
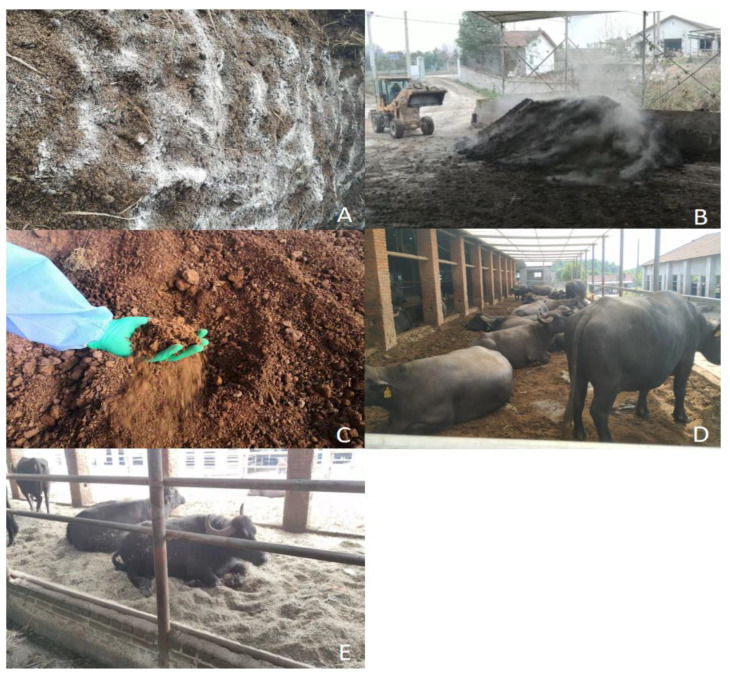
The production process of harmless fermentation bedding material of buffalo manure. (**A**) Collect fresh cow dung, adjust the moisture content and add microbial agent. (**B**) Turning the pile regularly. (**C**) Fermented manure bedding material. (**D**) Spread the FMB over the experimental group barn. (**E**) Spread the CB over the control group barn.

**Figure 2 animals-13-00842-f002:**
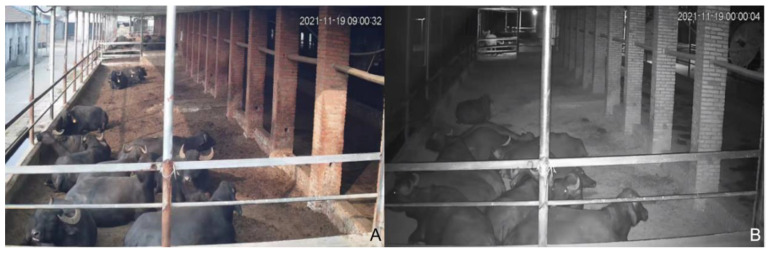
Daytime barn conditions (**A**), night barn conditions (**B**).

**Table 1 animals-13-00842-t001:** The factors influencing lying time of buffaloes and their interaction.

Dependent Variable	SS	df	MS	F	*p*-Value
Month	194,168.13	11	17,651.65	28.76	*p* < 0.001
Bedding	190,519.56	1	190,519.56	310.43	*p* < 0.001
Month * Bedding	16,864.05	11	1533.10	2.50	*p* < 0.01
Error	117,834.22	192	613.72		
Total	519,385.96	215			

Abbreviations: SS, sum of squares; df, degree of freedom; MS, mean square; *, interaction effect.

**Table 2 animals-13-00842-t002:** Average daily lying time of buffaloes with different bedding and month.

Month	Environment Temp (°C)	Average Daily Lying Time (min)		*p*-Value
		CB	FMB	
January	3.6–11.6	603 ± 4.20	646 ± 7.35	*p* < 0.01
February	5.0–13.1	614 ± 4.39	658 ± 9.70	*p* < 0.01
March	7.9–16.7	624 ± 5.13	693 ± 12.17	*p* < 0.01
April	12.4–18.5	633 ± 5.23	722 ± 9.79	*p* < 0.01
May	17.5–24.1	646 ± 6.75	713 ± 10.79	*p* < 0.01
June	22.4–32.0	614 ± 6.09	658 ± 12.29	*p* < 0.01
July	25.0–32.4	610 ± 6.03	632 ± 9.79	*p* = 0.068
August	23.3–29.9	542 ± 6.29	615 ± 8.34	*p* < 0.01
September	21.5–32.3	550 ± 8.22	616 ± 10.32	*p* < 0.01
October	14.2–21.2	599 ± 8.32	674 ± 10.70	*p* < 0.01
November	3.6–11.6	613 ± 5.42	678 ± 6.75	*p* < 0.01
December	3.2–11.2	619 ± 5.05	673 ± 10.20	*p* < 0.01

Abbreviations: CB, chaff bedding; FMB, fermented manure bedding.

**Table 3 animals-13-00842-t003:** The factors influencing standing time of buffaloes and their interaction.

Dependent Variable	SS	df	MS	F	*p*-Value
Month	31,282.72	11	2843.88	10.48	*p* < 0.001
Bedding	49,111.34	1	49,111.34	181.01	*p* < 0.001
Month * Bedding	5795.61	11	526.87	1.94	*p* < 0.05
Error	52,094.22	192	271.32		
Total	138,283.89	215			

Abbreviations: SS, sum of squares; df, degree of freedom; MS, mean square; *, interaction effect.

**Table 4 animals-13-00842-t004:** Average daily standing time of buffaloes with different bedding and month.

Month	Environment Temp (°C)	Average Daily Standing Time (min)		*p*-Value
		CB	FMB	
January	3.6–11.6	355 ± 6.43	331 ± 6.41	*p <* 0.05
February	5.0–13.1	347 ± 4.98	327 ± 9.83	*p* = 0.097
March	7.9–16.7	350 ± 3.99	312 ± 3.14	*p <* 0.01
April	12.4–18.5	346 ± 3.62	307 ± 3.54	*p <* 0.01
May	17.5–24.1	327 ± 3.50	308 ± 4.37	*p <* 0.01
June	22.4–32.0	350 ± 4.91	335 ± 8.30	*p* = 0.13
July	25.0–32.4	361 ± 6.31	341 ± 5.79	*p <* 0.05
August	23.3–29.9	378 ± 2.21	340 ± 4.69	*p <* 0.01
September	21.5–32.3	371 ± 5.39	342 ± 7.88	*p <* 0.01
October	14.2–21.2	362 ± 7.29	317 ± 4.08	*p <* 0.01
November	3.6–11.6	355 ± 3.83	310 ± 3.85	*p <* 0.01
December	3.2–11.2	343 ± 6.40	312 ± 3.89	*p <* 0.01

Abbreviations: CB, chaff bedding; FMB, fermented manure bedding.

**Table 5 animals-13-00842-t005:** BCI of buffaloes with different bedding.

Month	Environment Temp (°C)	BCI a.m. (%)		*p*-Value	BCI p.m. (%)		*p*-Value
		CB	FMB		CB	FMB	
January	3.6–11.6	87.78% ± 1.25%	90.07% ± 1.70%	*p* = 0.30	85.50% ± 1.79%	86.28% ± 1.38%	*p* = 0.73
February	5.0–13.1	87.40% ± 1.48%	91.21% ± 2.28%	*p* = 0.18	86.28% ± 1.38%	88.58% ± 2.03%	*p* = 0.36
March	7.9–16.7	86.91% ± 1.70%	92.06% ± 1.95%	*p* = 0.064	86.80% ± 1.64%	89.80% ± 1.65%	*p* = 0.22
April	12.4–18.5	86.84% ± 1.30%	89.07% ± 1.92%	*p* = 0.35	87.26% ± 1.71%	89.72% ± 1.24%	*p* = 0.26
May	17.5–24.1	87.14% ± 2.12%	87.29% ± 1.98%	*p* = 0.95	86.75% ± 1.68%	89.21% ± 1.32%	*p* = 0.26
June	22.4–32.0	86.56% ± 1.84%	90.16% ± 2.01%	*p* = 0.21	85.15% ± 0.87%	88.31% ± 1.56%	*p* = 0.09
July	25.0–32.4	82.53% ± 1.06%	86.91% ± 1.75%	*p <* 0.05	84.94% ± 1.76%	86.17% ± 1.23%	*p* = 0.57
August	23.3–29.9	85.16% ± 1.62%	87.78% ± 1.48%	*p* = 0.25	84.43% ± 1.25%	85.37% ± 1.34%	*p* = 0.62
September	21.5–32.3	86.52% ± 1.17%	89.52% ± 2.01%	*p* = 0.22	84.61% ± 1.62%	87.34% ± 1.83%	*p* = 0.28
October	14.2–21.2	86.93% ± 1.30%	89.88% ± 2.01%	*p* = 0.24	85.64% ± 1.01%	90.40% ± 1.48%	*p <* 0.05
November	3.6–11.6	86.83% ± 1.43%	89.74% ± 1.13%	*p* = 0.13	86.06% ± 1.84%	89.20% ± 1.24%	*p* = 0.17
December	3.2–11.2	88.11% ± 1.51%	89.74% ± 1.13%	*p* = 0.41	86.49% ± 1.52%	89.64% ± 1.51%	*p* = 0.16

Abbreviations: CB, chaff bedding; FMB, fermented manure bedding.

**Table 6 animals-13-00842-t006:** Average daily milk yield of buffaloes with different bedding materials.

Month	Environment Temp (°C)	Average Daily Milk Yield (kg)		*p*-Value
		CB	FMB	
January	3.6–11.6	4.29 ± 0.082	4.57 ± 0.062	*p <* 0.01
February	5.0–13.1	4.24 ± 0.060	4.51 ± 0.061	*p <* 0.01
March	7.9–16.7	4.39 ± 0.047	4.56 ± 0.074	*p <* 0.05
April	12.4–18.5	4.27 ± 0.044	4.51 ± 0.053	*p <* 0.01
May	17.5–24.1	4.22 ± 0.063	4.52 ± 0.054	*p <* 0.01
June	22.4–32.0	4.13 ± 0.075	4.24 ± 0.071	*p* = 0.28
July	25.0–32.4	3.80 ± 0.048	4.05 ± 0.053	*p <* 0.01
August	23.3–29.9	3.90 ± 0.071	4.13 ± 0.056	*p <* 0.05
September	21.5–32.3	3.75 ± 0.065	4.05 ± 0.048	*p <* 0.01
October	14.2–21.2	4.26 ± 0.041	4.43 ± 0.072	*p <* 0.05
November	3.6–11.6	4.27 ± 0.056	4.52 ± 0.043	*p <* 0.01
December	3.2–11.2	4.25 ± 0.059	4.54 ± 0.061	*p <* 0.01

Abbreviations: CB, chaff bedding; FMB, fermented manure bedding.

**Table 7 animals-13-00842-t007:** Body hygiene score of buffaloes with different bedding materials.

Month	Environment Temp (°C)	Body Hygiene Score		*p*-Value	Proportion of Dirty Buffalo		SEM	*p*-Value
		CB	FMB		CB	FMB		
January	3.6–11.6	1.75 ± 0.19	1.86 ± 0.20	*p* = 0.70	25.00%	33.33%	1.08%	*p <* 0.01
February	5.0–13.1	1.85 ± 0.20	1.81 ± 0.19	*p* = 0.16	30.00%	28.57%		
March	7.9–16.7	1.80 ± 0.19	1.95 ± 0.19	*p* = 0.57	25.00%	33.33%		
April	12.4–18.5	1.75 ± 0.18	1.86 ± 0.19	*p* = 0.69	20.00%	28.57%		
May	17.5–24.1	1.70 ± 0.19	1.90 ± 0.19	*p* = 0.46	25.00%	33.33%		
June	22.4–32.0	1.85 ± 0.18	1.81 ± 0.18	*p* = 0.87	25.00%	23.81%		
July	25.0–32.4	1.86 ± 0.17	1.96 ± 0.17	*p* = 0.69	23.81%	30.43%		
August	23.3–29.9	1.86 ± 0.19	2.00 ± 0.18	*p* = 0.58	28.57%	34.78%		
September	21.5–32.3	1.86 ± 0.19	1.87 ± 0.18	*p* = 0.96	28.57%	30.43%		
October	14.2–21.2	1.76 ± 0.18	1.87 ± 0.17	*p* = 0.67	23.81%	26.09%		
November	3.6–11.6	1.81 ± 0.16	1.91 ± 0.17	*p* = 0.66	19.05%	26.09%		
December	3.2–11.2	1.86 ± 0.17	1.91 ± 0.18	*p* = 0.82	23.81%	30.43%		

Abbreviations: SEM, overall standard error on the mean; CB, chaff bedding; FMB, fermented manure bedding.

**Table 8 animals-13-00842-t008:** Locomotion score of buffaloes with different bedding materials.

Month	Environment Temp (°C)	Locomotion Score		*p*-Value
		CB	FMB	
January	3.6–11.6	1.40 ± 0.11	1.48 ± 0.11	*p* = 0.63
February	5.0–13.1	1.45 ± 0.11	1.43 ± 0.11	*p* = 0.89
March	7.9–16.7	1.35 ± 0.10	1.43 ± 0.11	*p* = 0.62
April	12.4–18.5	1.45 ± 0.11	1.48 ± 0.11	*p* = 0.87
May	17.5–24.1	1.35 ± 0.11	1.43 ± 0.11	*p* = 0.62
June	22.4–32.0	1.40 ± 0.11	1.48 ± 0.11	*p* = 0.63
July	25.0–32.4	1.38 ± 0.11	1.43 ± 0.10	*p* = 0.72
August	23.3–29.9	1.48 ± 0.11	1.52 ± 0.11	*p* = 0.77
September	21.5–32.3	1.43 ± 0.11	1.48 ± 0.11	*p* = 0.75
October	14.2–21.2	1.48 ± 0.11	1.43 ± 0.11	*p* = 0.79
November	3.6–11.6	1.43 ± 0.11	1.39 ± 0.11	*p* = 0.81
December	3.2–11.2	1.38 ± 0.11	1.43 ± 0.11	*p* = 0.72

Abbreviations: CB, chaff bedding; FMB, fermented manure bedding.

**Table 9 animals-13-00842-t009:** Hock lesion score of buffaloes with different bedding materials.

Month	Environment Temp (°C)	Hock Lesion Score		*p*-Value
		CB	FMB	
January	3.6–11.6	0.85 ± 0.18	0.86 ± 0.17	*p =* 0.97
February	5.0–13.1	0.85 ± 0.19	0.90 ± 0.18	*p =* 0.84
March	7.9–16.7	0.95 ± 0.18	0.90 ± 0.18	*p =* 0.86
April	12.4–18.5	0.80 ± 0.17	0.86 ± 0.18	*p =* 0.82
May	17.5–24.1	0.90 ± 0.19	0.86 ± 0.17	*p =* 0.87
June	22.4–32.0	0.80 ± 0.19	0.90 ± 0.18	*p =* 0.69
July	25.0–32.4	0.86 ± 0.18	0.91 ± 0.18	*p =* 0.83
August	23.3–29.9	0.90 ± 0.18	0.87 ± 0.18	*p =* 0.89
September	21.5–32.3	0.81 ± 0.19	0.96 ± 0.17	*p =* 0.57
October	14.2–21.2	0.95 ± 0.17	0.91 ± 0.17	*p =* 0.88
November	3.6–11.6	0.86 ± 0.18	0.87 ± 0.18	*p =* 0.96
December	3.2–11.2	0.90 ± 0.17	0.96 ± 0.17	*p =* 0.83

Abbreviations: CB, chaff bedding; FMB, fermented manure bedding.

## Data Availability

The datasets used and or analyzed during the current study are available from the corresponding author.

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
