# Peer review of "Effects of Different Bedding Materials on Production Performance, Lying Behavior and Welfare of Dairy Buffaloes"

_animals, 2023, doi:10.3390/ani13050842_

Round 1
Reviewer 1 Report
General Comment
The article by Niuab et al. evaluated the utilization of fermented manure as a bedding material for buffaloes in a loose housing system. The ectopic fermentation method was employed to recycle manure for bedding. The study has practical implications for farmers raising buffaloes and is noteworthy. However, the manuscript requires significant revisions and language editing to enhance its overall quality. Specific comments and suggestion are listed below.
Summary
It is advisable to refrain from repeating similar information in different sentences.
Abstract
The abstract was fine.
Line 31: Was it idle standing in the pen?
Introduction:
L 43: is this information for general dairy farms in the world, or for a specific area?
Please add information on recycled manure and concise the information on other bedding materials. Add studies that compared recycled manure with other bedding materials. Additionally, some details about ectopic fermentation of manure would improve the quality of introduction. The authors are also encouraged to add some text about bedding materials being used in buffalo farming. It is very important to establish the need of the study.
Material and methods
Line 68: Did fermented manure include rice chaff as well? If so, please describe the ratio of manure and rice chaff.
Line 89: Any specific microbial agent?
Line 91: Add specific location of pile from where the temperature was recorded i.e. at what depth?
Line 92: For how many days this cycle would go?
Line 95: Any details about the rice chaff bedding?
Please add information on temperature and humidity of the area where this fermentation was done. These are important details to repeat the study at some other place.
Line 103: i.e. experimental group having bedding of fermented manure and control group with rice chaff bedding.
Line 105: The shed in not a freestall. It seems a naturally ventilated shed having loose housing with concrete floor covered with bedding material. Please correct the housing system.
Line 107: These two were the pens not the stalls. A stall is usually an elevated area for individual animal divided by partitions.
Line 109: Exercise area means the adjacent outdoor loafing area? Do add per buffalo space allowance.
Line 111: Please be specific. Add the frequency of adding the bedding material. Was it weekly, biweekly, or monthly?
Line 112: The description of monitoring lying behavior is great.
Line 125: Was it just the aimless standing? It appears it did not include walking time, or the time spent at the feed bunk or drinking. Please elaborate it.
Line 132: Was it machine milking or hand milking?
Line 140: Did the authors record body condition score as well?
Statistical Analysis
The observations were collected over a period of one year on same buffaloes. Therefore, the repeated measures ANOVA would be a correct way to analyze the data. Please revise the analysis. The line 141 described that analysis was done on group averages. It should be done on animal values not the averages of the groups.
There is very limited information about the buffaloes enrolled in the study. Please add the name of buffalo breed, average days in lactation at the start of the trial, average milk yield of each group at the start, and pregnancy status.
Results
Line 160: The table 2 clearly indicates a distinct temperature variation across different seasons. The authors are encouraged to have analysis on seasonal basis instead of months. The moths with similar average temperature can be grouped in corresponding season. Do consider the humidity as while categorizing the seasons.
Line 174: Please elaborate about standing posture. Was this the idle standing in the pen?
Discussion
This section needs improvement. The authors are requested to focus on the findings of their stud. For example, if the lying time increased on fermented bedding, provide logical reasoning for happening that. It is suggested to limit the number of comparisons to previous studies to one or two instances for each variable, as the emphasis should be on the current study's results and significance. Additionally, avoid repeating the complete results in discussion section.
Conclusion
Please limit your findings to the buffalo farming system.
Author Response
Response to Reviewer 1 Comments
A minor revision, following all the comments present in the annexed text, has to be made
Comments
Point 1: Abstract
Line 31: Was it idle standing in the pen?
Response 1: Thank you so much for your careful check.
The average daily standing time of the buffalo in this study was the time that the buffalo was supported by at least three legs on the ground, excluding the time when buffalo running, frolicking and roughhouse time.
Point 2: Introduction
Line 43: is this information for general dairy farms in the world, or for a specific area?
Please add information on recycled manure and concise the information on other bedding materials. Add studies that compared recycled manure with other bedding materials. Additionally, some details about ectopic fermentation of manure would improve the quality of introduction. The authors are also encouraged to add some text about bedding materials being used in buffalo farming. It is very important to establish the need of the study.
Response 2: Thank you so much for your careful check.
The most used bedding materials in current barns are wood chips and sand [1-2]. Other materials including straw, peanut shells are also often used [3-4]. This is information about the general dairy farms in the world.
We have improved it.
Point 3: Material and methods
Line 68: Did fermented manure include rice chaff as well? If so, please describe the ratio of manure and rice chaff.
Line 89: Any specific microbial agent?
Line 91: Add specific location of pile from where the temperature was recorded i.e. at what depth?
Line 92: For how many days this cycle would go?
Line 95: Any details about the rice chaff bedding?
Please add information on temperature and humidity of the area where this fermentation was done. These are important details to repeat the study at some other place.
Line 103: i.e. experimental group having bedding of fermented manure and control group with rice chaff bedding.
Line 105: The shed in not a freestall. It seems a naturally ventilated shed having loose housing with concrete floor covered with bedding material. Please correct the housing system.
Line 107: These two were the pens not the stalls. A stall is usually an elevated area for individual animal divided by partitions.
Line 109: Exercise area means the adjacent outdoor loafing area? Do add per buffalo space allowance.
Line 111: Please be specific. Add the frequency of adding the bedding material. Was it weekly, biweekly, or monthly?
Line 112: The description of monitoring lying behavior is great.
Line 125: Was it just the aimless standing? It appears it did not include walking time, or the time spent at the feed bunk or drinking. Please elaborate it.
Line 132: Was it machine milking or hand milking?
Line 140: Did the authors record body condition score as well?
Response 3: Thank you so much for your careful check.
Line 68: The raw material of FMB was buffaloes’ manure, in which rice chaff was added to adjust the moisture content.
Line 89: The microbial agent mainly contained bacillus subtilis DK068, lactic acid bacteria, yeast MX0126, cellulase, and lignin enzyme, with viable bacteria count > 1.0 × 1010 CFU/g.
Line 91-92: When the middle temperature (50cm) of the pile reaches 55℃, turn the pile with forklift every 4 days thereafter. The fermentation period is about 40 days.
Line 95: We have explained in the article
Line 103: Yes, fermented manure bedding was used in the experimental group and chaff bedding was used in the control group.
Line 111: Both stalls were supplemented with the appropriate bedding periodically every two months.
Line 125: The average daily standing time of the buffalo in this study was the time that the buffalo was supported by at least three legs on the ground, including the time when the buffalo eating food and drinking water, and excluding the time when the buffalo running, frolicking and roughhouse time.
Line 132: It's machine milking.
Line 140: All buffalo were in good and healthy condition during the study. We recorded buffaloes welfare indexes including body hygiene, locomotion and hock lesion scores.
Line 105-109: We attached a sketch of the structure of the cowshed. Do I need to add it to the text?We have attached a schematic diagram to describe the structure of the cowshed. Do I need to add it to the text?
Figure-1 Barn structure (1: Fixed ceiling; 2: Feeding channel; 3: Water trough; 4: CB bedding; 5: FMB bedding; 6: Ventilation window; 7: Enclosure fence; 8: Guardrail door; 9: Fan)
Point 4: Statistical Analysis
The observations were collected over a period of one year on same buffaloes. Therefore, the repeated measures ANOVA would be a correct way to analyze the data. Please revise the analysis. The line 141 described that analysis was done on group averages. It should be done on animal values not the averages of the groups.
There is very limited information about the buffaloes enrolled in the study. Please add the name of buffalo breed, average days in lactation at the start of the trial, average milk yield of each group at the start, and pregnancy status.
Response 4: Thank you so much for your careful check.
This study analyzed the production performance, lying behavior indexes and animal welfare indexes of buffaloes, and were conducted by two-way ANOVA in SPSS. p < 0.05 was used to indicate a significant difference.
All the buffaloes in this study were lactating Mediterranean hybrids, and all were in early or early-middle lactation stage.
Point 5: Results
Line 160: The table 2 clearly indicates a distinct temperature variation across different seasons. The authors are encouraged to have analysis on seasonal basis instead of months. The moths with similar average temperature can be grouped in corresponding season. Do consider the humidity as while categorizing the seasons.
Line 174: Please elaborate about standing posture. Was this the idle standing in the pen?
Response 5: Thank you so much for your careful check.
We still hope to analyze the ADLT and ADST of buffalo on the basis of month to reflect the influence of different bedding materials on the lying behavior of buffalo.
The average daily standing time of the buffalo in this study was the time that the buffalo was supported by at least three legs on the ground, excluding the time when the water buffalo running, frolicking and roughhouse time.
Point 6: Discussion
This section needs improvement. The authors are requested to focus on the findings of their stud. For example, if the lying time increased on fermented bedding, provide logical reasoning for happening that. It is suggested to limit the number of comparisons to previous studies to one or two instances for each variable, as the emphasis should be on the current study's results and significance. Additionally, avoid repeating the complete results in discussion section.
Response 6: Thank you so much for your careful check. We have modified it.
Point 7: Conclusion
Please limit your findings to the buffalo farming system.
Response 7: Thank you so much for your careful check. We have modified it.

Reviewer 2 Report
A few comments to attend to:
Line 80: Give coordinates of site of study
Line 81: Ethics Committee not Ethic
Line88: Explain a bit how you adjusted the moisture conytent of the manure
Line 89: What is the microbial agent?
Line 95: Indicate Chaff Bedding as CB and subsequently use the acronym CB
Line 102 - 103: State exact number of buffaloes used and not range (41 044, 21-23)
Line 111: State how often you supplemented bedding
Line 114: Indicate HD for High-definition cameras on first mention
Author Response
Response to Reviewer 2 Comments
A minor revision, following all the comments present in the annexed text, has to be made
Comments
Point 1: Line 80: Give coordinates of site of study
Response 1: Thank you so much for your careful check.
This study was conducted from December 2020 to December 2021 in Jinniu Animal Husbandry Co LTD, Jingmen city, Hubei province, China (Longitude: 112.07438; latitude: 30.72365).
Point 2: Line 81: Ethics Committee not Ethic
Response 2: Thank you so much for your careful check.
We have revised it from“ Ethic Committee ” to “ Ethics Committee ”.
Point 3: Line88: Explain a bit how you adjusted the moisture content of the manure
Response 3: Thank you so much for your careful check.
Firstly, fresh buffalo manure and rice chaff were collected. The moisture content of the chaff was less than 5%, and the moisture content of the mixture was adjusted to about 60% by adding chaff into the buffalo manure and mixing it evenly with a forklift.
Point 4: Line 89: What is the microbial agent?
Response 4: Thank you so much for your careful check.
The microbial agent mainly contained bacillus subtilis DK068, lactic acid bacteria, yeast MX0126, cellulase, and lignin enzyme, with viable bacteria count >1.0 × 1010 CFU/g.
Point 5: Line 95: Indicate Chaff Bedding as CB and subsequently use the acronym CB
Response 5: Thank you so much for your careful check.
Point 6: Line 102 - 103: State exact number of buffaloes used and not range (41 044, 21-23)
Response 6: Thank you so much for your careful check.
More than 40 multiparous lactating buffaloes were randomly divided into 2 groups according to age. There were 20 lactating buffaloes in FMB group and 21 in CB group from January to June. There were 21 lactating buffaloes in FMB group and 23 in CB group from July to December.
Point 7: Line 111: State how often you supplemented bedding
Response 7: Thank you so much for your careful check.
Both stalls were supplemented with the appropriate bedding periodically every two months.
Point 8: Line 114: Indicate HD for High-definition cameras on first mention
Response 8: Thank you so much for your careful check.
Reviewer 3 Report
I had enjoyed reading this very well written manuscript, and I want to congratulate the authors.
Bedding material and resting behaviour were shown to influence production efficiency and welfare of dairy cows, however, there is a substantial lack of knowledge on how these factors influence dairy water buffaloes. The experimental design looks solid to me, and also the methods applied.
I have some minor suggestions for the improvement of the manuscript, please see bellow:
Lines 102-103: Regarding the number of animals, please clary and give the exact number, I don’t understand how it is possible to give ranges for both number of buffalo cows from the control and experimental groups (please give the exact number of the final number of animals from which you collected data/group). And after, replace in the abstract section as well, with the final number;
Lines 110-111: How frequent, e.g. on a weekly basis/daily? Please give more detail. And also, could you provide a rough estimate for bedding/square meter or animal used?
Lines 114-116: Please give the commercial name of the video-cameras used and 2-3 technical parameters for each equipment.
Lines 212-2014: I am in agreement with the authors here, however, a more detailed explanation is needed here. Why do the buffalo cows stand more during heat stress? There are results showing that they adapt this behavior (standing perpendicular on the direction of wind) to help thermoregulation.
Lines 237-242: Please convert the yuan in international currency as well (American dollars or/and EURO), in order for readers to better grasp the findings of the current study. Also in the conclusions section, add the international currency.
Although, I am not an English native speaker, I noticed some syntax errors in the manuscript, please crosscheck for spelling the manuscript before resubmitting it.
Author Response
Response to Reviewer 3 Comments
A minor revision, following all the comments present in the annexed text, has to be made
Comments
Point 1: Lines 102-103: Regarding the number of animals, please clary and give the exact number, I don’t understand how it is possible to give ranges for both number of buffalo cows from the control and experimental groups (please give the exact number of the final number of animals from which you collected data/group). And after, replace in the abstract section as well, with the final number;
Response 1: Thank you so much for your careful check.
More than 40 multiparous lactating buffaloes were randomly divided into 2 groups according to age. There were 20 lactating buffaloes in FMB group and 21 in CB group from January to June. There were 21 lactating buffaloes in FMB group and 23 in CB group from July to December.
Point 2: Lines 110-111: How frequent, e.g. on a weekly basis/daily? Please give more detail. And also, could you provide a rough estimate for bedding/square meter or animal used?
Response 2: Thank you so much for your careful check.
Both stalls were supplemented with the appropriate bedding periodically every two months.
Point 3: Lines 114-116: Please give the commercial name of the video-cameras used and 2-3 technical parameters for each equipment.
Response 3: Thank you so much for your careful check.
The HD cameras and video recorders we used are the products of Zhejiang Dahua Technology Co., LTD. Model: DH-IPC-HFW1235M-12-V2. The video recorder contains four Seagate 4T solid state drives.
Point 4: Lines 212-2014: I am in agreement with the authors here, however, a more detailed explanation is needed here. Why do the buffalo cows stand more during heat stress? There are results showing that they adapt this behavior (standing perpen dicular on the direction of wind) to help thermoregulation.
Response 4: Thank you so much for your careful check.
We explain these results in the discussion section. Studies have shown that the more severe the heat shock, the lower the average daily lying time and frequency of cows. Cow core body temperature rises when lying down and decreases when standing.
Point 5: Lines 237-242: Please convert the yuan in international currency as well (American dollars or/and EURO), in order for readers to better grasp the findings of the current study. Also in the conclusions section, add the international currency.
Response 5: Thank you so much for your careful check.
We have converted the RMB in the results and conclusions into the international currency USD.

Round 2
Reviewer 1 Report
The authors have effectively addressed the comments in the manuscript. Only a minor suggestion is to delete the word "freestall" from the line 136 to avoid confusion about housing system. There is no need to include a farm sketch in the text as the figures effectively illustrate the information. Overall, the quality of the manuscript has been greatly improved.
Author Response
Response to Reviewer 1 Comments
A minor revision, following all the comments present in the annexed text, has to be made
Comments
Point 1: The authors have effectively addressed the comments in the manuscript. Only a minor suggestion is to delete the word "freestall" from the line 136 to avoid confusion about housing system. There is no need to include a farm sketch in the text as the figures effectively illustrate the information. Overall, the quality of the manuscript has been greatly improved.
Response 1: Thank you so much for your careful check. We have removed the word "freestall" from line 136, and thank you very much for your advice.
